# Dynamic Reconstruction of Hand-Object Interaction with Distributed Force-aware Contact Representation

## Abstract

We present ViTaM-D, a novel visual-tactile framework for dynamic hand-object interaction reconstruction, integrating distributed tactile sensing for more accurate contact modeling. While existing methods focus primarily on visual inputs, they struggle with capturing detailed contact interactions such as object deformation. Our approach leverages distributed tactile sensors to address this limitation by introducing DF-Field. This distributed force-aware contact representation models both kinetic and potential energy in hand-object interaction. ViTaM-D first reconstructs hand-object interactions using a visual-only network, VDT-Net, and then refines contact details through a force-aware optimization (FO) process, enhancing object deformation modeling. To benchmark our approach, we introduce the HOT dataset, which features 600 sequences of hand-object interactions, including deformable objects, built in a high-precision simulation environment. Extensive experiments on both the DexYCB and HOT datasets demonstrate significant improvements in accuracy over previous state-of-the-art methods such as gSDF and HOTrack. Our results highlight the superior performance of ViTaM-D in both rigid and deformable object reconstruction, as well as the effectiveness of DF-Field in refining hand poses. This work offers a comprehensive solution to dynamic hand-object interaction reconstruction by seamlessly integrating visual and tactile data. Codes, models, and datasets will be available.

## 1 Introduction

Human manipulates objects with tactile feedback. Reconstructing the manipulation process of hand-object interaction is an important task that can benefit many downstream tasks, such as VR/AR, robot imitation learning, and human behavior understanding. Previous works (Yang et al., 2021; 2024; Chen et al., 2023a; Tsoli & Argyros, 2018; Hasson et al., 2019b) on dynamic hand-object interaction reconstruction are primarily visual-only. Visual-only methods can recover the global geometry and poses of hand and object, but they struggle with contact details such as object deformation due to a lack of information near contact areas.

Recently, visual-tactile methods (Xu et al., 2023; Smith et al., 2020; Wang et al., 2018) have increasingly drawn attention in hand-object reconstruction tasks, with the development of tactile sensing techniques (Sundaram et al., 2019; Liu et al., 2017; Yuan et al., 2017; Ren et al., 2023), which can supplement the perceptual ability near contact areas (Fig. 1.a). Among the fast-developing tactile sensors, distributed tactile sensors (Fig. 1.b) (Liu et al., 2017; Yin et al., 2023; Sundaram et al., 2019), which are wearable or attachable to a hand and allow collecting tactile information in a natural human-like hand-object interaction process, are more promising in human manipulation data collection than other types of tactile sensors (Taylor et al., 2022; Yuan et al., 2017; Lambeta et al., 2020; Ren et al., 2023). However, how to integrate such sensors with visual perception to reconstruct the states of hand-object interaction is seldom explored.

Due to the conformality between the distributed tactile sensor and the hand, the hand geometry will not be significantly influenced during manipulation. Therefore, we can use the same visual-only approach to reconstruct the hand-object state and integrate the tactile information to refine the result. Based on this observation, we propose a novel **D**istributed **F**orce-aware contact representation, **DF-**

Figure 1: (a): The relationship between tactile information and contact geometry. Grasping different bowls with the same hand poses shows that the distributed tactile arrays can capture occlusion contacts and object states. (b): Different types of distributed tactile sensors proposed in previous works (1). Liu et al. (2017), (2). Yin et al. (2023), (3). Sundaram et al. (2019).

**Field**, and a **Vi**sual-**Ta**ctile **M**anipulation reconstruction framework with **D**istributed tactile sensing, **ViTaM-D**. DF-Field models the contact by considering both kinetic and potential energy in hand-object interaction, which allows modeling object deformation. With this representation, ViTaM-D first reconstructs the hand-object interaction with visual-only observations by the proposed **V**isual **D**ynamic **T**racking network, **VDT-Net**, and refines the contact details with DF-Field via a **F**orce-aware hand-pose **O**ptimization process, **FO**. Such design allows us to inherit the wisdom of fast-growing visual-only hand-object reconstruction methods (Yang et al., 2024; Chen et al., 2023a; Tsoli & Argyros, 2018; Hasson et al., 2019b) and easily integrate the tactile information into an existing motion capture or estimation system.

To train the VDT-Net, we need large-scale hand-object interaction datasets since previous datasets (Chao et al., 2021; Yang et al., 2022b; Fan et al., 2023) of hand-object interactions mostly cover rigid or articulated object manipulation, failing to contain deformable objects. And they generally do not provide accurate tactile readings. Therefore, aside from the public dataset, DexYCB (Chao et al., 2021), which we adopt to benchmark on rigid objects with common baselines, we also create a new dataset, **HOT dataset**, to fully benchmark our method on deformable object reconstruction. The dataset is built with ZeMa (Du et al., 2024), a high-precision physics-based simulation environment that supports penetration-free frictional contact modeling with finite element method (FEM) to model the object deformation. The HOT dataset contains 600 sequences of hand-object interaction, with 30 deformable objects from 5 different categories and 8 camera views for each sequence.

To evaluate the method, we compare our proposed ViTaM-D with previous state-of-the-art methods gSDF (Chen et al., 2023b) and HOTrack (Chen et al., 2023a) on DexYCB. Extensive experiments proved that our method realizes great improvements in the previous works both quantitatively and qualitatively. Besides, we also prove the great capability of our method in tracking deformable objects on the HOT Dataset and the effectiveness of the FO optimization in refining hand poses from penetrations and bad contact states.

Our contributions are summarized as follows:

(1) A visual-tactile learning framework, ViTaM-D. It contains a visual-only dynamic tracking network, VDT-Net, for reconstructing hand-object interactions and a force-aware optimization process, FO, to integrate the tactile information into reconstruction refinement based on a novel distributed force-aware contact representation, DF-Field.

(2) A new dataset, HOT. It contains 600 RGB-D manipulation sequences on 30 deformable objects from 5 categories with penetration-free hand-object poses and accurate tactile information.

## 2 RELATED WORK

Hand-object reconstruction has been richly studied because it tries to recover the full details of hand-object interaction, showing potential applicability for downstream tasks.

Research in this direction starts with static reconstruction. Earlier works have predominantly relied on RGB inputs to estimate hand pose and object geometry. Hasson et al. (Hasson et al., 2019a) presented a method that learns hand-object interaction using synthetic RGB images with the grasp poses planned by GraspIt! (Miller & Allen, 2004). Doosti et al. (Doosti et al., 2020) proposed a graph-based network for hand-object pose estimation. Cao et al. (Cao et al., 2021) introduced a

method that operates in-the-wild, estimating the hand pose first and optimizing the solution using a contact representation. The optimization involves push and pull terms to handle the contact, which is purely empirical. Similarly, CPF (Yang et al., 2021; 2024) employs a contact representation using a spring-mass system, adopting an empirical approach to refine the hand-object interaction. ArtiBoost (Yang et al., 2022a) further enhanced the performance of static hand-object reconstruction through data augmentation techniques. AlignSDF (Chen et al., 2022) leveraged RGB inputs to estimate hand pose and combined point cloud data for object geometry, using SDF-based decoders for both hand and object reconstruction.

Later, hand-object reconstruction in the dynamic setting draws increasing attention since manipulation is naturally dynamic. Approaches have advanced by incorporating temporal information and using more complex representations. Tekin et al. (Tekin et al., 2019) adopted RNN for temporal feature fusion, utilizing egocentric RGB video to capture dynamic hand-object interaction and pose estimation. Hasson et al. (Hasson et al., 2020) improved dynamic reconstruction by enforcing photometric consistency constraints. Ye et al. (Ye et al., 2023) introduced a diffusion model that guides dynamic hand-object reconstruction via score distillation sampling and differentiable rendering techniques. Recent advances by Fan et al. (Fan et al., 2024) focused on refining pose estimation using SDF-based representations and volumetric rendering, improving the overall dynamic reconstruction accuracy. Furthermore, gSDF (Chen et al., 2023b) employed transformer architectures and SDF representations to model complex hand-object interactions dynamically.

These works focus more on visual-only inputs. However, due to occlusion between the hand and object during the interaction, visual perception usually lacks information near the contact areas, and such information cannot necessarily be mitigated by cross-frame feature fusion. Therefore, tactile perception comes to supplement the near-contact information. Zhang et al. (Zhang et al., 2021) utilized a tactile glove (Sundaram et al., 2019) to track object movements. However, it focuses more on the dynamic object trajectory rather than contact geometry. Works such as (Smith et al., 2020; Wang et al., 2018) employed optical tactile sensors, trained the model by synthetic data, and applied to rigid object geometry reconstruction. Later, VTacO (Xu et al., 2023) extended this line of research by using optical tactile sensors to capture object geometry, including deformation, providing a more comprehensive representation of hand-object interactions. Unlike these works, the proposed ViTaM-D inherits the merits from both worlds; it leverages the advanced techniques of visual-only reconstruction and incorporates the distributed tactile readings to enhance the local contact details.

## 3 FORCE-AWARE CONTACT REPRESENTATION

During manipulation, forces from the hand change both hand and object states. To fully capture dynamic contact behaviors, the representation must encode both contact locations and forces. We introduce a distributed force-aware contact representation, **DF-Field** (Sec. 3.1), and apply it using distributed tactile sensors (Sec. 3.2).

### 3.1 DF-FIELD REPRESENTATION

Without loss of generality, we take the object-centric perspective to describe a hand-object interaction process, where the object is assumed to be fixed in the origin point, and the hand moves around the object. In this way, by ignoring the gravitational potential energy, the contact dynamics in a manipulation process are driven by the **Relative Potential Energy**, which is the resulting energy of the kinetic energy and deformation potential energy of the hand and object. Besides, we assume a virtual **Barrier Energy** lies between the hand and object contact surface to prevent penetration. With an established point pair $i, j$ of object vertex $\mathcal{V}_i^o$ and hand vertex $\mathcal{V}_j^h$, and the Euclidean distance of the point pair $l_{ij} = \left\| \mathcal{V}_i^o - \mathcal{V}_j^h \right\|_2$, we define the two energy terms as follows.

**Relative Potential Energy.** A relative potential energy that describes both the object deformation and the hand movements:

$$E_{ij} = \kappa l_{ij}^2, \tag{1}$$

where $\kappa$ is a parameter representing the hand interacting with the object vertices. If $\kappa > 0$, the hand and object are in contact. Thus, the distance $l_{ij}$ and the relative energy will be close to 0, indicating that the contact's relative potential energy is satisfied.

**Barrier Energy.** Given a certain threshold distance $\hat{l}$, the barrier energy is:

$$B_{ij} = \begin{cases} -e^{-\kappa}(l_{ij} - \hat{l})^2 \log\left(\frac{l_{ij}}{\hat{l}}\right), & 0 < l_{ij} < \hat{l} \\ 0 & l_{ij} \geq \hat{l} \end{cases} \tag{2}$$

This barrier energy aims to push away the point pair when $\kappa$ is low, thus avoiding penetration issues between the hand and object. The function is defined in this way not only to ensure repulsion becomes smaller when the distance is larger but also to remain smooth for optimization.

**Overall Energy.** A proper contact is met when both energy terms approach 0:

$$E = \sum_i \sum_j (E_{ij} + B_{ij}). \tag{3}$$

To note, though $\kappa$ is strongly correlated to the tactile readings, DF-Field can also be tactile-independent, as long as we empirically set the exerted force instead of physics-based. In this way, DF-Field can work with visual-only methods. Different setups of force are discussed in Sec. 6.5.

### 3.2 DF-FIELD WITH DISTRIBUTED TACTILE SENSORS

As aforementioned, the distributed tactile sensors generally conform to the hand, and different tactile sensors may have different layout configurations. Thus, to adapt to different tactile sensors, we take the hand-centric perspective by first dividing the hand into 22 regions (as shown in the left of Fig. 3.a, 2 areas for the thumb, 3 for other fingers, and 8 for the palm), and assign the tactile sensors to the corresponding region. In each region, we define the center as the hand keypoint, resulting in $\mathcal{K}^h \in \mathbb{R}^{22 \times 3}$.

For optimization, we connect only the hand keypoints to object vertices, reducing computational load while enhancing regional interaction information. The force in each region is calculated by averaging the tactile readings $\mathcal{M}^j$, and by the definition of the energy, we can obtain that $\kappa$ is the difference of the exerted force over the distance $l_{ij}$. In practice, we approximate it by dividing the force by the distance:

$$\kappa_{ij} \sim \frac{\overline{\mathcal{M}^j}}{l_{ij}}. \tag{4}$$

We will discuss the difference between choosing keypoints or all-hand vertices for optimization in Sec. Appendix B.2.

## 4 ViTaM-D METHOD

### 4.1 OVERVIEW

We address the problem of 4D tracking and dynamic reconstruction of hand-object interactions from a visual-tactile perspective and propose a novel method named **ViTaM-D**. We assume that the visual input consists of a live stream of 3D point clouds, denoted as $\{\mathcal{P}_t \in \mathbb{R}^{N \times 3}\}_{t=1}^n$, representing hand-object interactions derived from single-view depth images, where $N$ is the number of input point. The tactile input $\{\mathcal{M}_t\}_{t=1}^n$ contains distributed tactile sensor readings for each frame.

**ViTaM-D** tracks hand-object states in two stages: Visual Dynamic Tracking (**VDT-Net**) and Force-aware Optimization (**FO**). **VDT-Net** uses visual-only inputs to establish flow features and dynamically reconstruct the hand and object, capturing global hand-object geometry but struggling with sparse contact region details. To address this, the second stage, **FO**, incorporates tactile information to improve contact accuracy and reduce penetration issues. The overall pipeline is given in Fig. 2.

### 4.2 VISUAL DYNAMIC TRACKING OF HAND AND OBJECT, VDT-NET

In the first stage, the VDT-Net first predicts the flow from the last to the current frame with a flow prediction module and extracts a fused visual feature from the current frame's flow and point cloud. This feature will then be forwarded into **Object Decoder** and **Hand Decoder** to reconstruct the

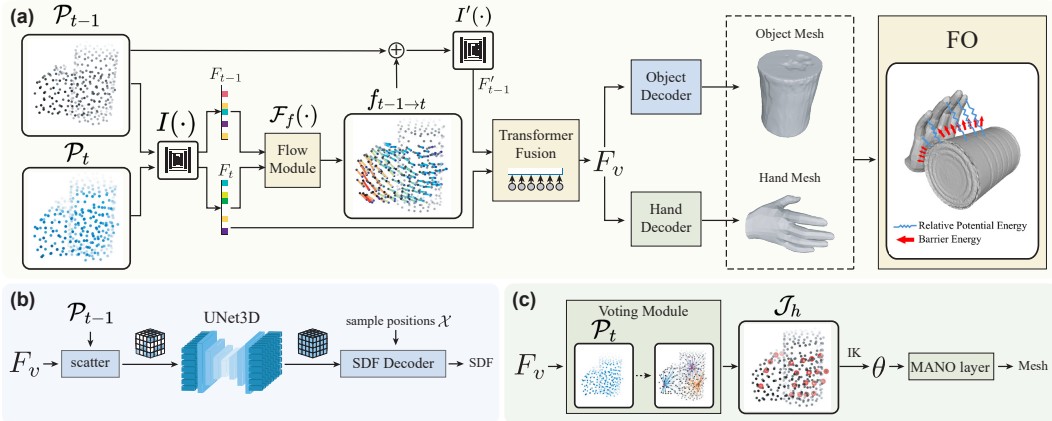

Figure 2: **ViTaM-D** pipeline. (a): The overview of our pipeline, including the flow prediction module for visual feature extraction and flow estimation, the Hand and Object Decoders, and Force-aware hand-pose Optimization, FO. (b): The Object Decoder to reconstruct the object mesh. (c): The Hand Decoder for estimating hand parameters and reconstructing based on the MANO model.

object and hand geometry, respectively. We use the Signed Distance Field (SDF) to model the objects and the MANO model (Romero et al., 2017) for the hand.

**Flow Prediction Module.** At frame $t$, we first extract the per-point features $F_t, F_{t-1} \in \mathbb{R}^{N \times d}$ from $\mathcal{P}_t$ and $\mathcal{P}_{t-1}$ using a backbone network $I(\cdot)$. In practice, we adopt a simple PointNet++ (Qi et al., 2017) for feature extraction, with 3 layers of set abstraction and 3 layers of feature propagation. Then, we design a flow prediction network $\mathcal{F}_f(\cdot)$ to predict the point cloud flow $f_{t-1 \to t} \in \mathbb{R}^{N \times 3}$ from frame $t - 1$ to $t$, which contains the correspondence information between the two frames and represents the hand movement and object deformation:

$$f_{t-1 \to t} = \mathcal{F}_f(F_t, F_{t-1}, \mathcal{P}_t, \mathcal{P}_{t-1}). \tag{5}$$

In $\mathcal{F}_f$, we first perform a Cartesian product of the two extracted features, yielding a tensor of size $N \times N \times 2d$. This tensor is fed into a 3-layer MLP to obtain $\mathcal{C}_v$, which is then used in two ways. First, it passes through a 2D convolutional layer for downsampling to obtain $p_c \in \mathbb{R}^{N \times N}$, representing the matching probability of each point between two frames. Second, $\mathcal{C}_v$ is sent through a softmax function and a one-layer MLP to downsample, resulting in $\mathcal{C}_c \in \mathbb{R}^N$, indicating whether the points in the first frame are matched in the second frame. Thus, the final matching probability matrix $p_m \in \mathbb{R}^{N \times N}$, which describes the correspondence likelihood between the two point sets, can be computed as:

$$p_m = p_c \times \mathcal{C}_c \tag{6}$$

After estimating the matching probability, we compute the disparity of two point cloud sets $\mathcal{D} \in \mathbb{R}^{N \times N \times 3}$, with $\mathcal{D}_{ij} = \mathcal{P}_t^i - \mathcal{P}_{t-1}^j$, and concatenate the disparity with $p_m$. The concatenated tensor is then fed into four 2D convolutional layers with batch normalization and a softmax function to obtain the disparity feature $F_d \in \mathbb{R}^{N \times d'}$. Finally, we use PointNet++ (Qi et al., 2017) to regress the flow $f_{t-1 \to t} \in \mathbb{R}^{N \times 3}$. The flow prediction results will be discussed in Appendix B.1.

Additionally, we use another PointNet++ $I'(\cdot)$ to extract the visual correspondence feature $F_t^f$ from $\mathcal{P}_{t-1}$ with $f_{t-1 \to t}$ added. $F_t^f$ and $F_t$ are then forwarded to a transformer fusion module to obtain the final visual feature $F_v$, corresponding to the current frame's point cloud $\mathcal{P}_t$. Specifically, the transformer fusion module first uses a self-attention module to encode the point cloud features from both frames. After applying positional embedding between the feature and its point cloud, a cross-attention module fuses the two features and outputs the final visual feature. This fusion strategy considers both the 3D static information of the current frame and the corresponding feature extracted from the flow.

**Object Decoder.** Based on the final visual feature $F_v$, the **Object Decoder** follows two steps: **Feature Scattering & Sampling**, and **SDF Decoding**, to obtain the SDF predictions and the object

mesh using Marching Cubes algorithm (Lorensen & Cline, 1987). We follow the design in ConvOc-cNet (Peng et al., 2020), which scatters the feature into volume with the resolution $D$, feeds it into a 3D-UNet, and finally uses a 5-layer MLP to predict the signed distance for every point.

**Hand Decoder.** For hand tracking and reconstruction, we adopt the parametric MANO hand model (Romero et al., 2017), which uses $\beta \in \mathbb{R}^{10}$ for hand shape, and $\theta \in \mathbb{R}^{51}$ for hand poses.

Based on the extracted final visual feature $F_v$, and the point cloud $\mathcal{P}_{t-1}$ of the last frame as input, we inherit a voting mechanism in PVN3D (He et al., 2020) to predict the hand joint positions $\mathcal{J}_h \in \mathbb{R}^{21 \times 3}$ of the current frame. Besides predicting the translation offset $\mathcal{O}_t \in \mathbb{R}^{21 \times 3}$, we also predict the initial position of the keypoint by estimating a probability matrix $T_{t-1} \in \mathbb{R}^{N \times 21}$ and multiply it with the former frame point cloud $\mathcal{P}_{t-1}$. The final prediction of the hand joint locations $\mathcal{J}_h$ is:

$$\mathcal{J}_h = \mathcal{O}_t + \mathcal{P}_{t-1} \times T_{t-1} \tag{7}$$

After obtaining hand joint locations $\mathcal{J}_h$, we use the inverse kinematics method (Zhou et al., 2020) to find the joint poses $\theta$ with a template hand shape $\beta$, and we can easily reconstruct the hand mesh through a differentiable MANO layer.

**Loss Function.** We train the network in an end-to-end way, and optimize the framework parameters with three loss terms:

$$\mathcal{L} = \lambda_f \mathcal{L}_{flow} + \lambda_S \mathcal{L}_{SDF} + \lambda_H \mathcal{L}_{Hand}, \tag{8}$$

where flow loss is defined by two Chamfer distance terms $\mathcal{L}_{flow} = CD(\mathcal{P}'_{t-1}, \mathcal{P}_t) + CD(\mathcal{P}''_t, \mathcal{P}_{t-1})$, $\mathcal{P}'_{t-1} = \mathcal{P}_{t-1} + f_{t-1 \to t}$ is the forward-shifted point cloud and $\mathcal{P}''_t = \mathcal{P}_t + f_{t \to t-1}$ is the backward-shifted point cloud. SDF loss is $\mathcal{L}_{SDF} = |s - s^*|$, where $s^*$ represents the ground truth of SDF. Hand joint loss is $\mathcal{L}_{Hand} = \|\mathcal{J}_h - \mathcal{J}_h^*\|_2^2$, with $\mathcal{J}_h^*$ the ground truth of hand keypoints.

### 4.3 FORCE-AWARE HAND-POSE OPTIMIZATION, FO

After proper training, the VDT-Net, as well as other visual-only approaches, can give decent outputs of hand-object reconstruction. However, information near the contact areas is lacking due to self-occlusion, and the contact details should be further improved. Here, we describe an optimization method with force readings.

Given the contact energy defined in Sec. 3, we optimize the predicted pose $\theta$ from the hand decoder, with respect to the reconstructed object mesh, to obtain a better hand mesh and contact state. Based on Equ. 3, we use the ball-query method to find the corresponding object vertices for each hand region with a radius $R$, and the point pair will be set up between them and the hand keypoints $\mathcal{J}_h$.

Besides, For a given joint $j$, we ensure it remains in reasonable poses by penalizing rotations $\mathcal{R}_j$ that are near twisted directions $\mathcal{R}_t$ or if any angle exceeds $\pi/2$, using the $L_2$ loss. Additionally, we constrain the optimized hand pose $^*\theta$ to stay close to the original prediction:

$$\mathcal{L}_r = \|\mathcal{R}_j \cdot \mathcal{R}_t\|_2^2 + \left\|\max(|\mathcal{R}_j| - \frac{\pi}{2}, 0)\right\|_2^2, \tag{9}$$
$$\mathcal{L}_o = \|^*\theta - \theta\|_2^2.$$

Finally, the optimization target can be demonstrated as:

$$^*\theta = \arg\min_{\theta}(E + \mathcal{L}_r + \mathcal{L}_o) \tag{10}$$

We use the gradient descent method and the Adam solver for optimization. By minimizing energy and loss, we can obtain a better interaction state between the hand and object, avoid severe penetration problems, and forbid the hand from abnormal poses.

## 5 HAND-OBJECT TACTILE DATASET, HOT DATASET

Since obtaining the ground truth in the real world, such as object deformation, is relatively hard, previous works (Chao et al., 2021; Yang et al., 2022b) on hand-object interaction recording mainly

focus on rigid or articulated objects. To fully benchmark our method's ability on deformable objects, we also create **H**and-**O**bject-**T**actile dataset, **HOT** dataset, which contains depth images, force sensor readings, and SDF ground truths generated by the simulation environment **ZeMa** (Du et al., 2024) which realizes intersection-free and high accuracy of contact modeling and deformation based on FEM.

## 5.1 DATA ACQUISITION

An example of collecting a training data sequence in the simulation environment is shown in Fig. 3.b. To ensure valid data points and stable grasps with multiple contacts between the objects and various regions of the hand, we use DiPGrasp (Xu et al., 2024), a fast and effective grasp planner. After obtaining a feasible grasp pose, we fix the selected object model, position the hand a certain distance from the final grasp translation, and add a slight perturbation to the final rotation. The hand is set fully open with joint states their lower limits. In the first stage, the hand moves slowly towards the final grasp transformation target. Next, we perform the grasp by gradually increasing the joint state values until they reach the grasp joint target. This process yields a complete training data sequence, including depth images, point clouds, and tactile arrays $\{\mathcal{P}_t, \mathcal{M}_t\}_{t=1}^n$ for each frame.

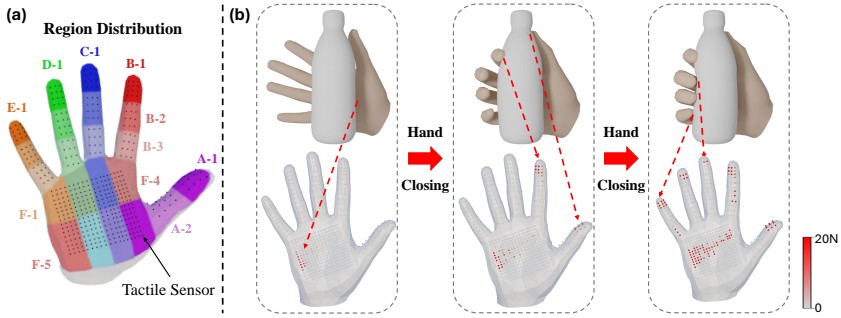

Figure 3: (a): The 22 regions and a typical distributed tactile sensor layout (Sundaram et al., 2019). (b): An example of collecting data in the simulation environment. For every frame, it generates depth images collected by different views of cameras and tactile sensor readings computed by ZeMa.

## 5.2 DATASET STATISTICS

In the **HOT** Dataset, we adopt objects from **YCB** (Calli et al., 2015) model repositories, which contain various object categories. We selected 5 to 10 different objects in the category **Bottle** and **Box** from the YCB dataset, and we manually added 5 **Sponges**, 5 **Plasticines**, and 5 **Stuffed Animals** to demonstrate the ability of our model to track objects with large deformations.

The simulator ZeMa mainly uses density $\rho$, Young's modulus $E$, and Poisson's ratio $\nu$ to describe the properties of different deformable objects. We will therefore use the parameters of objects that correspond to reality. For the object in category **Bottle** and **Box**, we set $\rho = 10^3 kg/m^3$, $E = 1.271$GPa, $\nu = 0.28$, and for the **Sponges**, **Plasticines** and **Stuffed Animals**, $\rho = 30 kg/m^3$, $E = 0.1$MPa, $\nu = 0.38$. And as we consider **Plasticine** as plastic objects, we set its yield stress $s_y = 200$pa.

For each object, we randomly generate 20 different trajectories, with 15 for training, 2 for testing, and 3 for validation and visualization. The whole dataset consists of 600 sequences which contains 50-100 frames, and each sequence involves 8 different views of camera for capturing point clouds.

## 6 EXPERIMENTS

## 6.1 IMPLEMENTATION DETAILS

We set the input point cloud size to $N = 1024$. We randomly sample $1 \times 10^6$ positions in the space, including $2 \times 10^5$ points on the object surface. By subsampling to $M = 2048$, we obtain the sample positions for SDF prediction during training. The point cloud feature size is set to $d = 128$,

the disparity feature size to $d' = 64$, and the resolution of the volume feature for SDF decoding to $D = 64$. We empirically set the loss weights to $\lambda_f = 0.01$, $\lambda_S = 0.5$, and $\lambda_H = 1$.

We first train our VDT-Net on the entire dataset with a batch size of 6 and a learning rate of $1e - 4$ using the Adam optimizer for 100 epochs. We then fine-tune each object category with a batch size of 4 and a learning rate of $5e - 5$ for 50-100 epochs, depending on the number of objects. The training process takes 15 hours on an Nvidia A40 GPU.

For the FO process, we set the threshold distance in the barrier function to $\hat{l} = 2$mm and the ball-query radius for finding object vertices to $R = 5$mm. We minimize the energy function for 100 iterations per frame with a learning rate of $2e - 3$ using the Adam solver. It takes approximately $3.5 \pm 0.5$ seconds per frame to complete refinement.

## 6.2 DATASETS

To train and evaluate our method, we first use the dataset **DexYCB**, which captures hand grasping of rigid objects, with RGB-D cameras recorded. It can directly validate our method's capability against the baselines. In addition, we use the HOT dataset, which contains interactions of hand and deformable objects with rich and accurate tactile information. It can evaluate the ability of our method to track object deformation and the effectiveness of our designed force-aware contact representation.

## 6.3 METRICS

**Intersection over Union (IoU)** evaluates the intersection percentage between the predicted object mesh and the ground truth.

**Chamfer Distance (CD)** assesses the reconstruction accuracy of object vertices.

**Mean Per Joint Position Error (MPJPE)** evaluates the accuracy of hand joint location tracking.

**Penetration Depth (PD)** measures the physical plausibility of hand-object interaction by reporting the maximum distance of hand penetration into the object.

**Hand-Object Contact Mask IoU (CIoU)** validates the contact recovery of hand-pose optimization by measuring the IoU of the contact mask between the optimized hand-object state and the ground truth. Contacts are defined as distances less than 3mm between hand and object vertices.

## 6.4 RESULTS

For hand-object tracking evaluation, we compare our results with the state-of-the-art methods gSDF (Chen et al., 2023b) and HOTrack (Chen et al., 2023a). gSDF uses RGB images to reconstruct objects with SDF representation, while HOTrack uses segmented point clouds to predict object poses. Our method uses unsegmented point clouds as input and output object meshes, making it more challenging to achieve better results. Additionally, since these baselines only work with rigid objects, we compare them only with the DexYCB object categories. DexYCB does not provide tactile information, so we only use VDT-Net to compare with the baseline methods here. Later, we provide an empirical way to use FO with a fixed force setting, which will be discussed in Sec. 6.5.

The quantitative results are shown in Tab. 1. In the DexYCB dataset, our leading scores in IoU and Chamfer distance demonstrate superior object tracking and reconstruction capabilities. Additionally, our higher MPJPE score indicates better hand-tracking performance. gSDF's better performance on penetration depth is due to occasional incomplete object reconstruction, as illustrated in Fig. 4, where the pitcher is missing its handle. For the HOT dataset, VDT-Net achieves excellent scores. Significant improvements in MPJPE, PD, and CIoU after applying our force-based optimization confirm the effectiveness of our design.

Fig. 4 shows the qualitative results of our method. Our tracking and reconstruction outperform state-of-the-art methods in the DexYCB dataset and show excellent performance on deformable objects in the HOT dataset. Our method accurately tracks hand movements and object deformations, thanks to the fusion of flow features and original point cloud features using a transformer, which incorporates both the 3D information of the current frame and correspondence features from the previous frame.

| Metrics | IoU(%)↑ | CD(mm)↓ | MPJPE(mm)↓ | PD(mm)↓ | CIoU(%)↑ |
|---|---|---|---|---|---|
| DexYCB | | | | | |
| gSDF | 86.8 | 13.4 | 14.4 | **8.9** | 31.3 |
| HOTrack | 88.2 | 10.2 | 25.7 | 12.3 | 28.5 |
| Ours (VDT) | **90.1** | **9.6** | **13.2** | 9.9 | **35.4** |
| HOT | | | | | |
| Ours (VDT) | 80.5 | 10.9 | 13.6 | 10.7 | 29.8 |
| Ours (VDT + FO) | * | * | **11.3** | **7.3** | **40.3** |

Table 1: Quantitative results on DexYCB and HOT datasets for previous SOTA and our method. We only compare our results on rigid objects with the baselines. * indicates that adding FO doesn't impact the metrics on reconstructed objects. ↑ / ↓ indicates higher scores/lower scores are better.

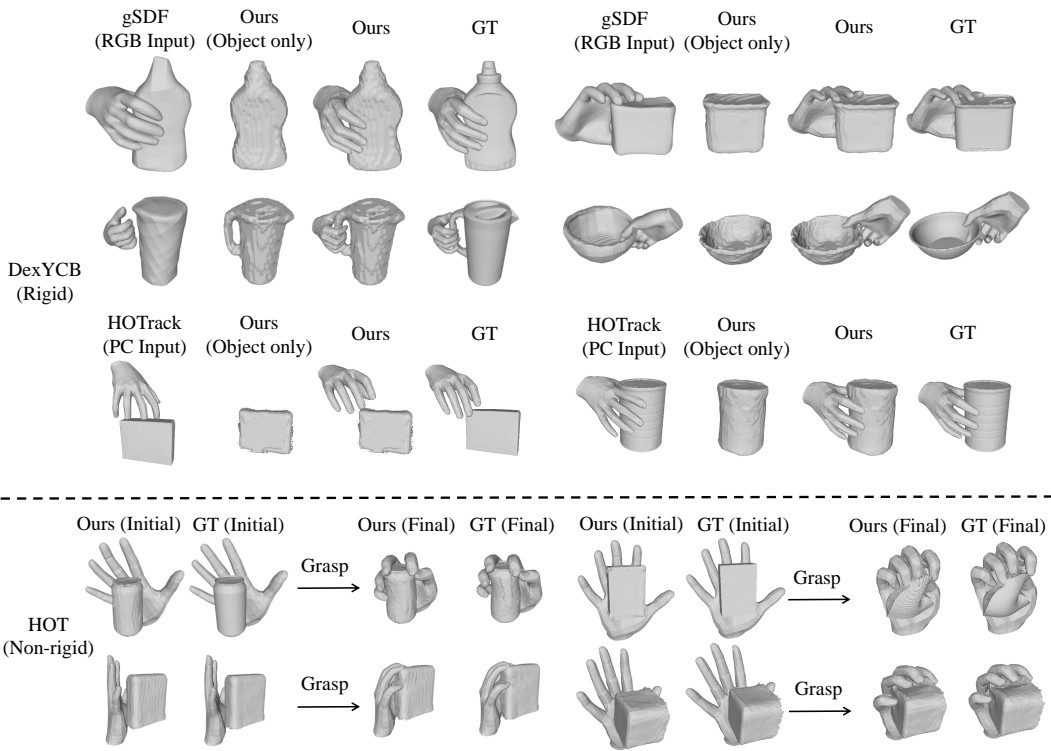

Figure 4: Qualitative results on both DexYCB and HOT datasets. The upper shows our better performances on rigid objects compared to the baselines, and the lower demonstrates the effectiveness of VDT-Net in dynamic reconstruction for deformable objects and hands.

Fig. 5 demonstrates the performances of force-aware hand-pose optimization. Due to the design of force-aware contact representation, the optimization can solve most of the penetration problems and refine the contact map based on collected tactile information.

## 6.5 ABLATION STUDY

We will conduct an ablation study on the following aspects: First, we demonstrate the results of introducing the distributed tactile arrays in the VDT-Net. Second, we discuss different definitions of force representation based on heuristic annotations or tactile-aware forms. In the appendix, we will also discuss the different point pair establishment methods in our contact representation and the importance of the flow prediction module.

Figure 5: Qualitative results with FO. As iteration steps rise, the penetration problem decreases, and the contact map is closer to the ground truth, making the contact more reasonable.

**Tactile feature fusion in VDT-Net.** To assess the impact of introducing distributed tactile arrays in VDT-Net, we first use a 3-layer MLP to encode the tactile features of each region. By estimating hand pose, we fuse these regional features to the sample points, adding the encoded tactile feature to the point-wise feature of each sample position. We train VDT-Net with fused tactile information on our HOT dataset, and the results are shown in Tab. 2. Quantitative results show no significant improvements, likely because the tactile data are much more sparse than the visual inputs, causing feature misalignment. Therefore, we implement DF-Field to convert force readings into contact states for hand-pose optimization.

| Metrics | IoU(%)↑ | CD(mm)↓ |
|---|---|---|
| VDT | 80.5 | **10.9** |
| VDT with Tactile | **81.2** | 11.5 |

Table 2: Quantitative results for whether or not fusing tactile information in VDT-Net.

| Metrics | MPJPE | PD | CIoU |
|---|---|---|---|
| DexYCB | | | |
| VDT | 13.2 | 9.9 | 35.4 |
| VDT+FO($\mathcal{M}$ fix) | **12.3** | **8.5** | **39.7** |
| HOT | | | |
| VDT+FO($\mathcal{M}$ fix) | 12.9 | 8.5 | 36.8 |
| VDT+FO | **11.3** | **7.3** | **40.3** |

Table 3: Quantitative results for different force representations.

**Different force representations.** We also discuss the effect of different force representations on our contact representation. For the DexYCB dataset, since no tactile information was provided, we test our hand-pose optimization process with the force reading $\mathcal{M}_j = 0.5$ for every region $j$ and compare the result with VDT-Net only. For the HOT dataset, we also conduct FO with $\mathcal{M}_j = 0.5$ for testing the influence of removing tactile information.

The results are shown in Tab. 3. We can see that on the DexYCB dataset, even with $\mathcal{M}$ fixed, the results are also improving in all aspects, which proves the great effectiveness of our force-aware contact representation. As for the results on the HOT dataset, we can conclude that introducing tactile information improves both penetration problems and contact recovery, mostly because the distributed tactile arrays provide accurate hand-object interaction forces, which is more reliable than setting force parameters empirically.

## 7 CONCLUSION

In this work, we introduce ViTaM-D, a dynamic hand-object interaction reconstruction framework that integrates distributed tactile sensing with visual perception. Featuring the DF-Field representation and force-aware optimization (FO), our approach effectively captures fine contact details and object deformation, and outperforms visual-only methods. We also present the HOT Dataset for benchmarking deformable object manipulation. Evaluations show that ViTaM-D outperforms state-of-the-art methods on both rigid and deformable objects. Future work includes integrating ViTaM-D with real tactile sensors and applying it to robot imitation learning, dexterous manipulation and human-robot collaboration.

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

## A  FLOW PREDICTION RESULT

To validate the accuracy of our proposed flow prediction module (FPM), we report the Chamfer distance error on the DexYCB and HOT datasets in Tab. 4. The relatively low Chamfer distance demonstrates the efficacy of our network. The slightly better results on the DexYCB dataset are likely due to the larger hand-object movements and more challenging object deformations in the HOT dataset.

We also present some qualitative results in Fig. 6. The top section compares our predicted flow added to the last frame's point cloud with the ground truth of the current frame's point cloud. The near overlap of the two point clouds indicates high prediction accuracy. The bottom section shows a sequence of our estimation results, illustrating our method's ability to track hand movements and object deformations in whole sequences.

| Dataset | DexYCB | | HOT | |
|---------|--------|--------|--------|-----------|
| Category | Box | Bottle | Sponge | Plasticine |
| CD(mm)↓ | 8.7 | 9.1 | 10.3 | 12.2 |

Table 4: Quantitative results for flow predictions in VDT-Net.

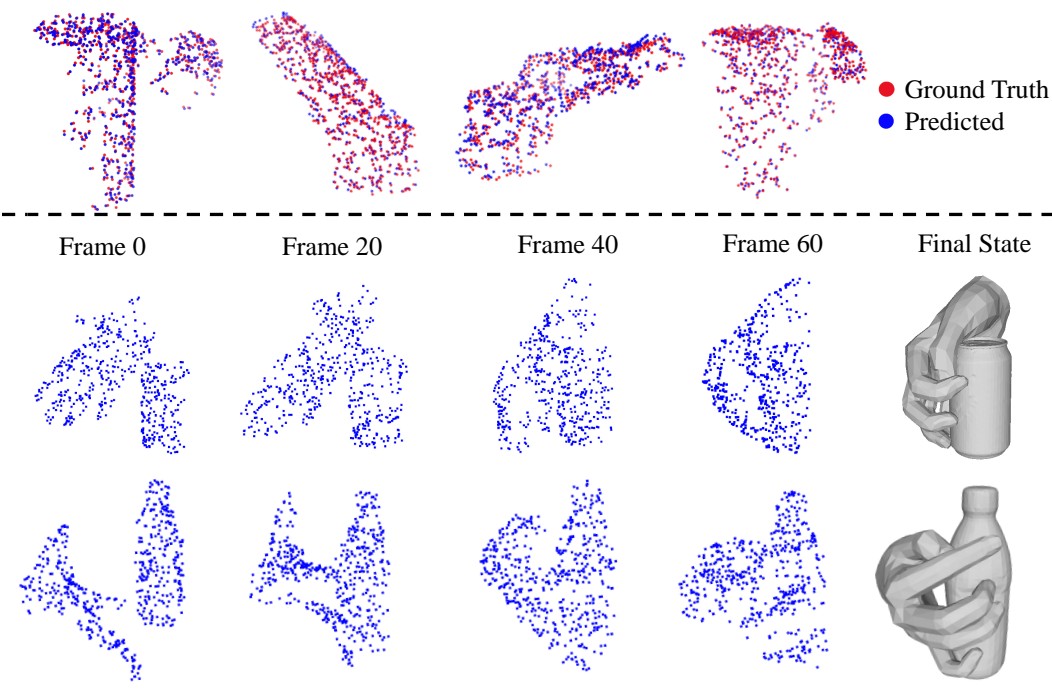

Figure 6: Qualitative results on flow prediction using FPM.

## B  ABLATION STUDY, EXTENDED

### B.1  IMPORTANCE OF FLOW PREDICTION MODULE

To demonstrate the importance of our flow prediction module (FPM), we conducted an experiment by directly fusing the visual features extracted from the input point clouds using the transformer fusion layer. The quantitative results are shown in Tab. 5. Introducing FPM significantly improves all scores, validating our feature fusion approach that incorporates the 3D static information of the current frame and the correspondence feature from the flow.

| Metrics | IoU(%)↑ | CD(mm)↓ | MPJPE(mm)↓ |
|---|---|---|---|
| Dataset | DexYCB | | |
| VDT(w/o FPM) | 84.3 | 15.7 | 17.1 |
| VDT | **90.1** | **9.6** | **13.2** |
| Dataset | HOT | | |
| VDT(w/o FPM) | 64.7 | 20.2 | 16.3 |
| VDT | **80.5** | **10.9** | **13.6** |

Table 5: Quantitative results on DexYCB and HOT Dataset for whether using the flow prediction module (denoted "FPM" in the table). "w/o" indicates without.

### B.2  DIFFERENT POINT PAIR ESTABLISHMENT STRATEGIES

This section discusses the influence of two point pair establishment strategies: using **keypoints** or **all-hand vertices**. When considering all hand vertices, we establish point pairs between them and nearby object vertices, treating the force exerted by the hand as the reading from the nearest sensor. The quantitative results of these two methods are shown in Tab. 6. While penetration depth improves slightly, both contact IoU and MPJPE decrease. This is likely because considering all

| Metrics | MPJPE(mm)↓ | PD(mm)↓ | CIoU(%)↑ | Iter. Time(s)↓ |
|---|---|---|---|---|
| Keypoint | **11.3** | 7.3 | **40.3** | **3.5 ± 0.5** |
| All Hand Vert. | 14.5 | **6.9** | 25.6 | 37 ± 3 |

Table 6: Quantitative results on evaluating point pair establishment on key points and on all-hand vertices.

sensors leads to conflicting optimization directions for the same joint, as sensors within the same regions may experience different contact situations. Additionally, the iteration time increases about tenfold compared to our setting.

