# OpenReview forum: "Dynamic Reconstruction of Hand-Object Interaction with Distributed Force-aware Contact Representation"
_ICLR.cc/2025/Conference — ICLR 2025 Conference Withdrawn Submission_

### Official Review · Reviewer_x7LG · 2024-10-28

**Soundness:** 3
**Presentation:** 3
**Contribution:** 2
**Rating:** 6
**Confidence:** 5

**Summary:**

This paper proposes a framework for hand-object reconstruction that integrates tactile information to model a more accurate contact. Authors first built upon VDT-Net that reconstructs hand-object interaction from visual input. Then, the reconstructed hand and object are refined by force-aware optimization (FO). Such optimization is based on DF-Field that leverages distributed tactile sensor data on top of visual data.

**Strengths:**

1. This paper proposes to integrate tactile data and tactile representation for modeling accurate contact, which is underexplored approach in hand-object reconstruction
2. The proposed method is easy to understand and well-written in terms of readability.
3. The VDT+FO achieves SOTA performance compared to RGB-based counterpart gSDF and point-based counterpart HOTrack.

**Weaknesses:**

1. Although the paper claims that it reconstructs hand and object from visual data, their visual input consists of streams of 3D point clouds (unlike RGB visual data from gSDF). This is hardly visual data, as most papers refer to visual data as RGB input data. Such terminology may, hence mislead many of the readers.
2. Despite multiple previous methods on how to effectively optimize hand and object based on novel contact representation such as ContactOpt (Grady et al., CVPR 2021), TOCH (Zhou et al., ECCV 2022), this paper did not compare force-aware optimization (FO) process with such methods.
3. While contact may exist outside regions of hand (dorsum side), tactile representation are only distributed within the inner side of the hand (palmar side). This may neglect contact from dorsum side of the hand. Such problem is not prevalent in previous works (e.g., ContactOpt, TOCH).
4. Minor mistakes: (1) “Learning joint reconstruction of hands and manipulated objects” is cited twice, (2) "Besides, For a given joint" should be "Besides, for a given joint" in L305.

**Questions:**

1. Why does the paper compare its method with single image-based hand-object reconstruction method (e.g., gSDF) while the paper addresses problem of 4D tracking with ViTaM-D. Maybe comparison with video-based methods like HOLD (Fan et al., CVPR 2024) may be more proper?
2. What is the reason that the method predicts flow from time t-1 to time t? If the model already possesses point cloud for time t, wouldn't it be more intuitive to directly predict 3D geometry based on the time t point input?
2. If the paper does not take any RGB input, wouldn't it be unfair comparison with gSDF? As reconstructing 3D hand and object directly from RGB data is much more difficult than reconstructing 3D hand and object from point cloud input, I believe further explanation on fairness of comparison should be discussed.
3. If gSDF re-implementation takes  point cloud as input, how is the implementation done? As gSDF is based on RGB image input, it seems non-trivial to implement gSDF with point cloud input.
4. How is the memory requirement of the proposed method compared with gSDF and HOTrack? Curious because the correspondence prediction seems to be memory-heavy.
5. Is it fair to fine-tune VDT-Net on each object category (from L382) while (as far as I know) other methods (e.g., gSDF) did not take such fine-tuning approach on each object category but rather trained the training set as a whole?
6. Is there any reason why the numbers for gSDF from this paper does not match that of Table 6 from gSDF's manuscript?
7. Is there a specific reason why implementing tactile sensor concept for contact representation is superior than modeling contact representation directly but naively based on distance between all vertices of hand mesh and all vertices of object mesh?
9. If previous contact-based hand-object optimization strategies are implemented instead of force-aware optimization (FO) process, would they underperform compared to FO process?
10. Current approach seems to exploit synthetic tactile data (HOT dataset) built upon ZeMa. Could this approach be extended to real tactile sensor data?

---

### Official Review · Reviewer_tXer · 2024-11-01

**Soundness:** 2
**Presentation:** 3
**Contribution:** 2
**Rating:** 3
**Confidence:** 4

**Summary:**

This work proposes a dynamic 3D hand-object interaction reconstruction framework ViTaM-D. To obtain more accurate hand poses during hand-object being in contact procedure, ViTaM-D firstly performs 3D hand-object shape reconstruction using point clouds with their VDT-Net, then further refines 3D hand poses using distributed tactile data. To achieve the optimization goal, this work introducing distributed force-aware contact representation, DF-Field, to model the hand-object contact. Since there lack of datasets, that meets the research needs, this work also creates a new synthesized dataset HOT, containing rigid and deformable objects, point clouds as well as tactile recordings. Finally, the paper presents the comparison experiments on DexYCB dataset with gSDF and HOTrack methods, as well as the results on HOT dataset.

**Strengths:**

1. This work focuses on using combined visual-tactile information to achieve more accurate hand-object 3D reconstruction, which is valuable but underexplored at present, especially when the grasped objects are deformable.
2. This work attempts to introduce new tactile related force representation DF-Field by leveraging kinetic and potential energy theory to model hand-object contact attributes and object deformation.

**Weaknesses:**

1. The definition of the tactile related force representation DF-Field seems somewhat rough. The presentations lack of detailed theoretical analysis from the perspective of physical laws, but directly define the formulas (1) and (2). That might lead to a weak support for the effective utilization of the tactile information.

2. The cascaded mode of fusing visual-tactile information lacks novelty, ignoring the real-time information fusion of the visual and tactile perception data. According to the provided definition and strategy, it seems that the tactile information might not be necessary, because the same refinement of hand 3D pose could also be obtained through geometric constraints. According to the analysis of results shown in Table 3, using fixed tactile recordings to all the contact regions also works well. So It might mean that it is not necessary to use tactile sensors. For the details, please refer to my questions.

3. Since this work adopts a cascading approach of first obtaining the 3D hand object shape through visual reconstruction and then further optimizing the hand pose, the performance of this proposed optimization approach FO should be examined on the basis of those reconstruction results using other SOTA visual models, such as gSDF. However, the paper does not provide experimental results in this regard. Furthermore, from a purely visual reconstruction performance comparison perspective, the proposed VDT-Net model uses point clouds as input, while the compared method gSDF uses monocular RGB as input, which has somewhat unequal data conditions.

**Questions:**

1. Question about force-aware hand-pose optimization. As presented in section 4.3, the adopted optimization loss formula (9) penalizes excessive rotation or large angular changes of joints using Lr, and keeps the optimized hand pose stay close to the original prediction using Lo. Both Lr and Lo reflect the geometric constraints. Only the loss E relates to tactile. According to formula (1) to (4), if the distance between the hand keypoint and the object vertex lij is larger than the threshold, it will result in Bij=0, therefore only the relative potential energy Eij will be preserved in the overall energy E. While Eij~Mlij, when the lij is large enough, M will equal to 0. In this situation, the hand pose optimization seems to have lost its driving force. That is to say, if the initial prediction of one joint wrongly stays away from the surface of an object(deep penetration or untouched), the proposed optimization method PO would not have the ability to correct it. So how to solve this problem? If the tactile readings are fixed, E will only proportional to geometric distance, irrelevant to tactile. How to explain? In addition, the commonly used penetration loss and attraction loss based on geometric distance of paired points have been proved effective to make hand-object contact better. What are the advantages of the proposed PO method compared to the commonly used method?

2. About the tactile annotations. As illustrated in Figure 3, the collected tactile information is annotated on the hand surface. In section 5.1, line 339, it says that the training data sequence includes depth images, point clouds, and tactile arrays for each frame. However, in Figure 5, the description is ‘… and the contact map is closer to the ground truth, making the contact more reasonable.’ It seems that there are ground truth contact maps annotated on the object surface. How to get these ground truth contact maps? How to use these ground truth contact maps during training the proposed model? What are the differences between the tactile annotations and the contact maps? Why not examine if the predicted tactile distribution on the hand regions is consistent with the ground truth tactile annotations?

---

### Official Review · Reviewer_Y4pP · 2024-11-03

**Soundness:** 3
**Presentation:** 1
**Contribution:** 2
**Rating:** 3
**Confidence:** 3

**Summary:**

The paper proposes a two-step pipeline to dynamically reconstruct hand-object interaction for hand/object models with Distributed Force-aware contact representation (DA-Field) and a Visual-Tactile Manipulation reconstruction framework (ViTaM). VDT-Net first reconstructs hand-object interaction with visual-only input, and use DA-Field to refine the contact detail. The paper also proposes a simulation dataset with both rigid and deforamble objects, with some extent of experiments with recent methods and ablations.

**Strengths:**

- The paper proposes a new dataset. Compared to existing work such as DexYCB which includes hand-object interaction with only rigid objects, the proposed HOT dataset includes both rigid and deformable objects, which is of potential value to the research community towards more general deformable objects.
- The paper seems to be in the right direction to model generic hand-object interaction that includes both rigid and deformable objects.
- Qualitative figures illustrate their points pretty well. For example, though the benchmark runs on simulator, Fig. 1 and Fig. 3 connect the proposed method to practical applications pretty well. Fig. 4 also provides many qualitative comparisons.

**Weaknesses:**

- **The writing, or the presentation, of the paper requires a significant amount of effort to meet the standard of an ICLR paper.** This is a major weakness in the current version of the paper. The writing makes it hard for the audience to understand what the technical contributions are. To enumerate a few points:
    - The authors proposed a confusing amount of acronyms in the paper. To name a few, DF-Field, ViTaM-D, VDT-Net, FO. Some of these acronyms are used abruptly without introducing the context first (e.g., DF-Field in the abstract), and some of these acronyms (e.g., FO) are only used a few times in the paper. The point of using acronyms is to make a particular component easier to read and memorize - the purpose of replacing a few occurrences of force-aware optimization with 'FO' seems to serve only the opposite.
    - The narration of the paper is centered around **what** the technical details are but not **why** these design choices are made and **how** they are motivated. This is bad because readers get lost in implementation details instead of getting insights that may be applicable to future research. For example, the second paragraph in the introduction section simply enumerates what existing work is doing, and ends with 'However, how to integrate such sensors with visual perception ... is seldom explored'. 'Seldom explored' implies that there *are* work that explores this. Instead of explaining how this paper differs from existing work and the novelty, the narration stops here, and begins to enumerate acronyms in the next paragraph. The method section also suffers from this issue. For instance, Sec. 3.1 introduces energy to propose hand-object interaction. Why is energy preferable than other types of modeling method?
- **Missing ablation.** Most of the design choices such as using energy to model the contact are not discussed. Without either conceptual justification or experimental justification, it is hard to assess the value of each proposed component; nor to find insights for related research. What part of the method contributes most to the improvement in Table. 1?
- **Missing baseline.** [A] seems to be a highly related work. Though it is cited in the paper, there are only a few sentences in the introduction and the related work sections to provide brief comparisons. More dicussion is needed.

[A] Xu, Wenqiang, et al. "Visual-tactile sensing for in-hand object reconstruction." Proceedings of the IEEE/CVF Conference on Computer Vision and Pattern Recognition. 2023.

**Questions:**

- Improvement of the presentation of the paper is needed (detailed above).
- What's the relationship between DA-Field and FO? From L066-L-074, it seems that DA-Field and FO are differnt things. Is FO a characteristic of DA-Field?
- Can you provide more ablations for each component? For example, Tab. 2 only ablates simple fusion of tactile features. The role of DA field is missing.
- SInce both DexYCB and HOT datasets are possible - would training on both datasets improve performance? Is sim2sim transfer possible?

In summary, my current evaluation leans towards the negative side primarily due to concerns on the presentation of the paper and unclear ablations.

---

### Official Review · Reviewer_UxYP · 2024-11-03

**Soundness:** 2
**Presentation:** 2
**Contribution:** 3
**Rating:** 6
**Confidence:** 3

**Summary:**

This paper introduces ViTaM-D, a novel framework integrating distributed tactile sensing into dynamic hand-object interaction reconstruction, aiming to improve contact modeling accuracy. The authors propose a Distributed Force-aware contact representation (DF-Field), which models both kinetic and potential energy in interactions. ViTaM-D employs a two-stage approach: an initial visual reconstruction using a dynamic network (VDT-Net), followed by force-aware optimization (FO) utilizing tactile readings to refine hand pose and contact accuracy. Additionally, the authors present the HOT dataset, which provides both visual data and tactile readings, including deformable objects, to benchmark this approach. The work is robust but could benefit from further comparison to other contact modeling-based hand pose refinement methods, as well as detailed explanations of practical applications involving both visual and tactile sensing. I recommend a weak accept based on its potential but encourage further elaboration on specific aspects.

**Strengths:**

1. **Novel Integration of Tactile Information**: By combining conventional visual reconstruction with tactile data, the authors effectively address limitations seen in purely visual methods, where contact precision often depends on inferred data. The use of tactile information in DF-Field provides direct contact modeling, introducing additional "knowledge" that improves interaction fidelity.
2. **Innovative DF-Field and Optimization**: The DF-Field approach, coupled with force-aware optimization, is both intuitive and innovative. The results demonstrate that DF-Field enhances reconstruction accuracy, especially in object deformation contexts, as supported by experiments. The HOT dataset, which includes tactile and deformable object data, further substantiates the framework's value.

**Weaknesses:**

1. **Clarification on Real-World Applicability**: The authors could enhance the paper by explaining the practical contexts where distributed tactile sensors would be available, and the types of real-world or robotic scenarios that could leverage both point cloud data and tactile sensing.
2. **Justifying Tactile Use Beyond Visual Information**: While the authors use tactile data to refine hand poses initially estimated by the network, it would strengthen the paper to add more experiments to compare with other spatial relationship-based hand pose refinement approaches (e.g., ContactOpt[1], TOCH[2], and S2Contact[3]), especially, the pose refinement part using contact.  Comparing tactile-based refinement with these methods could demonstrate the advantages of incorporating tactile data. Additionally, a comparison with VTacOH[4] regarding the methodology, performance and the usage of tactile data, could clarify the unique benefits of the proposed approach.

[1] Grady, P., Tang, C., Twigg, C. D., Vo, M., Brahmbhatt, S., & Kemp, C. C. Contactopt: Optimizing contact to improve grasps. CVPR 2021.

[2] Zhou, K., Bhatnagar, B. L., Lenssen, J. E., & Pons-Moll, G. Toch: Spatio-temporal object-to-hand correspondence for motion refinement. ECCV 2022.

[3] Tse, T. H. E., Zhang, Z., Kim, K. I., Leonardis, A., Zheng, F., & Chang, H. J. S2 contact: Graph-based network for 3d hand-object contact estimation with semi-supervised learning. ECCV 2022.

[4] Xu, W., Yu, Z., Xue, H., Ye, R., Yao, S., & Lu, C. Visual-tactile sensing for in-hand object reconstruction. CVPR 2023.

**Questions:**

**Q1**: In Table 1, could you provide more details on the baseline settings? Specifically, how were the baseline results obtained—did you retrain these models, or were pre-existing checkpoints used? Additionally, the Chamfer Distance (CD) values in Table 1 differ from those in Table 6 of the gSDF paper. Could you clarify the reasons for this discrepancy?

**Q2**: In Table 1, since object reconstructions are not optimized in your approach, could poor-quality object reconstructions lead to worse results after optimization? To help clarify this impact, could you present per-category results to show if object reconstruction quality would affect the final hand pose and contact performance after optimization?

**Q3**: In Figure 4, the reconstructed object mesh appears coarser than those in gSDF and HOT, even though you report using the same SDF resolution as gSDF and achieve a lower Chamfer Distance. Could you explain this discrepancy in visual quality?

**Q4**: The ablation study setup in Table 2 is somewhat unclear. I would expect the comparison of incorporating tactile information directly within the network (VDT with Tactile) versus using it in post-optimization (VDT+FO). Ideally, this setup would allow for a comparison of hand pose and contact outcomes between these approaches. However, only object reconstruction results are shown in Table 2. Could you include metrics such as MPJPE, PD, and CIoU to provide a more comprehensive evaluation and enable direct comparison with Table 1?

**Q5**: In Table 3, this ablation study is also unclear. From Figure 4, it appears that most contact occurs on the fingers, while the palm typically remains non-contact. Assigning a uniform force reading $M_j = 0.5$ across all regions is obviously wrong. Given that your initial result from VDT is refined using this inaccurate information yet still shows improvement, it’s difficult to derive meaningful insights from this ablation, as your experimental premise is wrong. Since Table 1 already demonstrates the effectiveness of force-aware optimization, could you clarify the motivation for this particular ablation study?

---

### Note · Authors · 2024-11-14

I have read and agree with the venue's withdrawal policy on behalf of myself and my co-authors.